

# Age of air from in situ trace gas measurements: Insights from a new technique

Eric A. Ray[1,2], Fred L. Moore[2,3], Hella Garny[4,5], Eric J. Hintsa[2,3], Bradley D. Hall[3], Geoff S.
Dutton[2,3], David Nance[2,3], James W. Elkins[3,6], Steven C. Wofsy[7], Jasna Pittman[7], Bruce Daube[7],
Bianca C. Baier[3], Jianghanyang Li[8,9] and Colm Sweeney[3]

[1]NOAA Chemical Sciences Laboratory, Boulder, CO, USA
[2]Cooperative Institute for Research in Environmental Science, CU Boulder, CO, USA
[3]NOAA Global Monitoring Laboratory, Boulder, CO, USA
[4]Deutsches Zentrum für Luft- und Raumfahrt (DLR), Institut für Physik der Atmosphäre, Oberpfaffenhofen,
Germany
[5]Ludwig-Maximilians-University Munich, Meteorological Institute, Munich, Germany
[6]Retired
[7]Department of Earth and Planetary Sciences, Harvard University, Cambridge, MA, USA
[8]Department of Atmospheric and Oceanic Sciences, CU Boulder, CO, USA
[9]Institute of Arctic and Alpine Research, CU Boulder, CO, USA

Correspondence to: Eric A. Ray, eric.ray@noaa.gov

**Abstract**. The age of air is an important transport diagnostic that can be derived from trace gas measurements and compared to global chemistry climate model output. We describe a new technique to calculate the age of air, measuring transport times from the Earth's surface to any location in the atmosphere based on simultaneous *in situ* measurements of multiple key long-lived trace gases. The primary benefits of this new technique include (1) optimized ages of air consistent with simultaneously measured $SF_6$ and $CO_2$, (2) age of air from the upper troposphere through the stratosphere, (3) estimates of the second moment of age spectra that have not been well constrained from measurements and (4) flexibility to be used with measurements across multiple instruments, platforms and decades. We demonstrate the technique on aircraft and balloon measurements from the 1990s, the last period of extensive stratospheric *in situ* sampling, and several recent missions from the 2020s, and compare the results with previously published and modeled values.

## 1 Introduction

Age of air has been used as an observationally-based diagnostic of atmospheric transport for a number of decades
(e.g., Bischof et al., 1985; Woodbridge et al., 1995; Schoeberl et al., 2005; Engel et al., 2017; Ray et al., 2022). The



stratospheric age of air is controlled by both the residual mean circulation and quasi-horizontal mixing (Waugh and Hall, 2002; Ploeger et al., 2014) and thus can give unique insight into these primary transport processes that govern the distribution of radiatively and chemically important trace gases in the stratosphere. An ideal trace gas to calculate mean ages, the first moment of the age spectra, would have a linear growth rate and be essentially inert in

the atmosphere. Since this ideal trace gas does not exist in nature, we use measured trace gases with long lifetimes and photochemical production or loss that can be accounted for and are in growth or decay. Most calculations of mean ages from *in situ* measurements have been based on a single trace gas with these characteristics, such as carbon dioxide ($CO_2$) (Bischof et al., 1985; Schmidt and Khedim, 1991; Woodbridge et al., 1995; Boering et al., 1996; Andrews et al., 2001) or sulfur hexafluoride ($SF_6$) (Volk et al.,1997; Engel et al., 2008). In the case of $CO_2$,

simultaneous measurements of methane ($CH_4$) and nitrous oxide ($N_2O$) are needed to adjust for $CO_2$ production and to account for the complication of interpreting the seasonal cycle of $CO_2$ for the youngest ages (Boering et al., 1996; Andrews et al., 2001).

Each age of air calculation technique from a single trace gas has complications that lead to uncertainty in the age

values. These complications include accurately accounting for photochemical production or loss of a trace gas, such as the mesospheric loss of $SF_6$ (Loeffel et al., 2022; Garny et al., 2024), or accurately characterizing the boundary condition of a trace gas, whether at the tropopause or the Earth's surface (e.g., Engel et al., 2008). Comparisons between mean age estimates from different trace gas measurements have been difficult due to the sparsity of simultaneous measurements of multiple age-of-air species, but when available the differences have been shown to be

significant (e.g., Leedham Elvidge et al., 2018). This uncertainty from the measurement-based estimates is compounded by the large disagreement among chemistry-climate model (CCM) and reanalysis-based age of air (Chabrillat et al., 2018; Ploeger et al., 2019; Abalos et al., 2021) that leaves us with a significant gap in our quantification of the stratospheric age of air and thus also the important processes that determine it as well as their changes over time.


One possibility to improve the calculation of age of air from observations is with the use of multiple simultaneous or near simultaneously measured trace gases, which has been shown previously to have potential for calculating age of air with more fidelity compared to estimates from a single trace gas (e.g., Schoeberl et al., 2005; Ehhalt et al., 2007; Boenisch et al., 2009; Ray et al., 2022). The main advantage of using multiple trace gases is to reduce the

uncertainties or biases caused by the characteristics of a single trace gas to define the age of air (e.g. the seasonal cycle of $CO_2$). A detail of these techniques is that each trace gas does not contribute equally to the determination of age of air, that is, only the longest-lived trace gases that are in growth (or decay) can quantify the range of ages in the stratosphere (Podglajen and Ploeger, 2019). Thus, trace gases such as $CO_2$, $SF_6$ and carbon tetrafluoride ($CF_4$), referred to as primary age trace gases, can be used to make independent estimates of the age, while shorter lived

trace gases such as trichlorofluoromethane (CFC-11) or chloromethane ($CH_3Cl$) can be used only in addition to the longer-lived trace gases to give information about the shortest ages (weeks to months) typically found in the tropopause region (Ray et al., 2022).



The main disadvantages of using multiple trace gases in an age of air calculation are (1) the previously mentioned
sparsity of simultaneous measurements of the most suitable trace gases and (2) the detailed knowledge of boundary
conditions for each trace gas that are required for accurate age of air estimates. The sparsity of simultaneous
measurements is due to the different measurement techniques and sampling frequency for each trace gas, as well as
aircraft and balloon payloads that may not have included multiple instruments suitable for age-of-air trace gas
measurements. The knowledge of boundary conditions for many trace gases, especially either at the tropical surface
or tropopause, is limited by whether long term measurements by a network are available. These disadvantages have
confined age of air estimates with multiple tracers to limited times and locations (e.g. Luo et al., 2018; Ray et al.,
2022).

In this work we demonstrate a new optimization technique to obtain mean age of air, along with several other
transport diagnostics, from simultaneous measurements of multiple primary age trace gases. The technique is
applicable to any primary age trace gases and we use the examples of $SF_6$ and $CO_2$ here. This method is somewhat
of a hybrid between the traditional single trace gas mean age estimates and the multiple trace gas techniques,
especially the work of Boenisch et al. (2009). As in Boenisch et al. (2009) and Ray et al. (2022), we leverage
additional information beyond mean age obtained in the optimization of a solution from multiple trace gases. Also,
similar to those two studies, we use the Earth's surface as the boundary condition of age of air, since that is where
we have long-term measurements of each trace gas. This method allows for transport characteristics of the upper
troposphere and tropopause region to be included as well as information about the latitudinal surface source regions.
The use of $SF_6$ measurements in the technique presented here is made possible by new estimates of $SF_6$ mesospheric
loss (Garny et al., 2024) that are used in the optimization.


This paper focuses on the details of the new technique with some example results using measurements from the
1990s and more recently from the 2020s. A follow up paper will explore the differences in the age of air between
these two time periods. The next section describes the data used in the examples shown in here. Section 3 describes
the method and Sect. 4 shows some results based on *in situ* measurements.


**2 Data**

For the stratospheric data in this study, we use aircraft and balloon-based *in situ* measurements of $SF_6$ (Elkins et al.,
1996; Moore et al., 2003; Hintsa et al., 2021), $CO_2$ (Daube et al., 2002), $CH_4$ and $N_2O$ (Webster et al., 1994, 2001;
Loewenstein et al., 2002) as well as balloon-based AirCore measurements of these four species (Baier et al., 2021;
Li et al., 2023). AirCores (or larger-volume, dual-dimension StratoCores) are measured by two different analytical
systems for trace gas species used in this analysis: by cavity ring-down spectrometry for $CO_2$, $N_2O$ and $CH_4$, and by
a gas chromatograph coupled with an electron capture detector for $SF_6$ and $N_2O$ (Li et al., 2023). We use
measurements from both the 1990s, the last time period of extensive stratospheric *in situ* sampling of age of air trace



gases above 16 km, and from several recent missions and routine measurements in the 2020s.  The 1990s ER-2
     aircraft missions include ASHOE-MAESA (1994), STRAT (1995-96), POLARIS (1997) and SOLVE (1999-2000)
     and balloon-based data from Observations of the Middle Stratosphere (OMS, 1996-2000).  The 2020s airborne
     missions include the Dynamics and Chemistry of the Summer Stratosphere (DCOTSS, 2021-22, ER-2) and
     Stratospheric Aerosol, processes, Budget and Radiative Effects (SABRE, 2023, WB-57) aircraft campaigns as well
as routine AirCore sampling (2021-24, Karion et al., 2010; Baier et al., 2021; Li et al., 2023).  Mean ages have been
     previously calculated from the measurements in the 1990s, which allows us to compare the results shown here with
     those values (Volk et al., 1997; Andrews et al., 2001), but no mean ages have yet been published based on the 2020s
     measurements.

In order to use the method described in the next section we need simultaneous measurements of several trace gases,
     which has been exceptionally rare even when the aircraft or balloon platform carried all of the required instruments,
     due to different sampling and calibration timing.  *In situ* $CO_2$, as well as $CH_4$ and $N_2O$, have typically been
     measured at ~1-5 second resolution, while $SF_6$ has been measured at ~1-2 minute resolution.  In order to markedly
     increase the number of sample times with simultaneous mole fractions of the four trace gases considered here, we
interpolated in time across measurement gaps of up to 100 seconds in one or more of the trace gases.  This time gap
     corresponds to a horizontal distance of ~20 km at typical ER-2 aircraft flying speeds and a vertical distance of ~0.3-
     0.5 km for typical balloon ascent and descent rates. For long-lived trace gases in the stratosphere such as those
     considered here, variability over the interpolated spatial scales is sufficiently small in almost all cases to not
     significantly affect the results.

     A critical part of the method is to have latitudinally and time-varying surface mole fractions of the key trace gases
     listed above.  For the surface boundary condition of each trace gas, we use the NOAA Greenhouse Gas Marine
     Boundary Layer (MBL) Reference product (https://gml.noaa.gov/ccgg/mbl/index.html; Lan et al., 2023) when
     available.  This is a latitudinally-resolved MBL product derived from weekly measurements of the four trace gases
considered in this study.  For time periods before the beginning of the NOAA MBL product we extend the time
     series based on different sources depending on the trace gas.  For $CO_2$ before 1979 and $N_2O$ before 2001, we use the
     CMIP6 reference time series from Meinshausen et al. (2017).  For $SF_6$ before 1997, we use scaled Cape Grim
     mixing ratios based on Engel et al. (2008), which are very similar to the CMIP6 reference values.  For $CH_4$ before
     1983, we use annual average mixing ratios from Etheridge et al. (1998).  For each trace gas we add on an average
seasonal cycle and latitudinal gradient based on the earliest available five years of the NOAA MBL product.

**3 Method**

The mixing ratio, $\chi$, of a trace gas, $i$, at a location, $\boldsymbol{x}$, and time, $t$, can be expressed as:



$$\chi_i(\boldsymbol{x},t) = \int_0^\infty \chi_{io}(t-t')e^{-t'/\tau_i(x,t')}G(\boldsymbol{x},t,t')dt', \tag{1}$$

where $G$ is the age spectrum, $t'$ is the transit time from a source region to the location $\boldsymbol{x}$, $\chi_{io}(t-t')$ is the mixing

ratio time series at the source region and $\tau_i$ is the path dependent lifetime of the trace gas (Fig. 1) (Schoeberl et al.,

2000; Ehhalt et al., 2007). The age spectra are assumed to have an inverse Gaussian functional form given by

$$G(\boldsymbol{x},t,t') = \sqrt{\frac{\Gamma^2}{4\pi R t'^3}}\, exp\left(\frac{-\Gamma^2(t'-\Gamma)^2}{4Rt'}\right) \tag{2}$$

where the mean age is given by $\Gamma = \int_0^\infty t'G(\boldsymbol{x},t,t')dt'$, the width of the spectrum is given by $\Delta^2 =$

$\frac{1}{2}\int_0^\infty (t'-\Gamma)^2 G(\boldsymbol{x},t,t')dt'$ and the ratio of moments $R = \Delta^2/\Gamma$ (Hall and Plumb, 1994; Waugh and Hall, 2002).

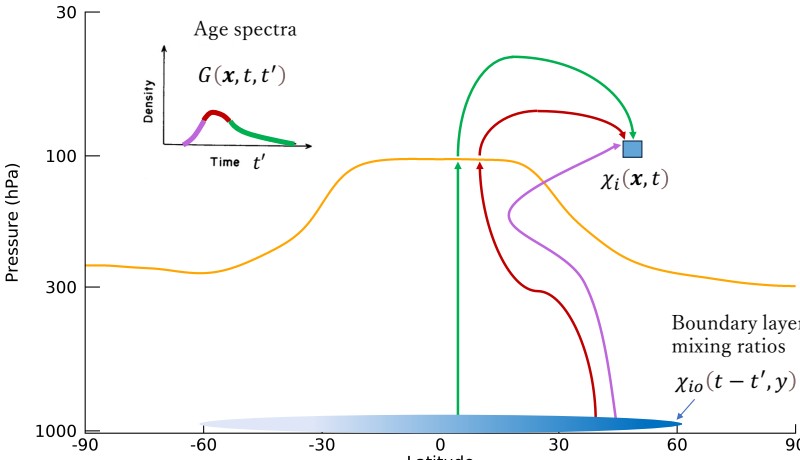

Figure 1. *Schematic of the age of air with the surface as the source region. Different pathways to the sampled*

*region ($\boldsymbol{x}$) are represented by the colored lines with different surface latitudinal origins (y) and time scales of*

*transport ($t'$) represented by the age spectra (G).*

The ratio of moments $R$, which is a measure of the width of the age spectra, is a largely unconstrained quantity and

so we allow a range of values from 0.1-2.5 years for each possible $\Gamma$ in the optimization calculation (Fig. S1). Most

studies of age of air have used values of $R$ ranging from 0.7-1.25 years based on model estimates (e.g., Hall and

Plumb, 1994; Volk et al., 1997; Engel et al., 2008). However, with a better understanding of the effect of the



exponential tail of $G$ for $t' > 10$ years, the model estimates of $R$ have increased to values of 1.5 years or more with considerable variability in the stratosphere (e.g., Diallo et al., 2012; Ploeger and Birner, 2016; Fritsch et al., 2020).

The source region for our calculation is the Earth's surface even though most previous stratospheric age of air studies have used the tropical tropopause since that is the primary entry region for air to the stratosphere above 380K (e.g., Andrews et al., 2001; Engel et al., 2009). There are a number of reasons we choose to use the surface as the source region: (1) we have long-term measurements of all the trace gases at the surface which is not the case for the tropical tropopause, (2) we can obtain information about the surface source latitude regions with our optimization

technique, (3) most modeled mean ages use the surface as the source region and (4) we can calculate ages in the troposphere and lowermost stratosphere with the same technique as the stratospheric overworld.

Following the techniques of Hauck et al. (2020) and Ray et al. (2022), we partition the age spectra into a part with a tropical (30°S-30°N average) surface source, $g_{TR}$, and a part with a latitudinally varying surface source, $g_V$,


$$G(\Gamma, \Delta, t') = g_{TR}(\Gamma, \Delta, t') + g_V(\Gamma, \Delta, t'). \tag{3}$$

The lowercase $g$ is used to indicate age spectra that are non-normalized, in contrast to the uppercase $G$ total age spectra that are normalized. The age spectra $g_V$ does not vary with latitude but is convolved with surface mixing ratio time series from varying latitudes as part of the optimization as will be shown in the next section, whereas the

age spectra $g_{TR}$ are always convolved with the tropical average surface mixing ratios. Note that we have now expressed the age spectra as a function of $\Gamma$ and $\Delta$ since those are the parameters in $G$ we allow to vary in the convolutions with the $\chi_{io}$ time series. The partitioned age spectra are expressed as fractions, $f$, of the $G$ as a function of $t'$.


$$g_{TR}(\Gamma, \Delta, t') = f(t')G(\Gamma, \Delta, t') \tag{4}$$

$$g_V(\Gamma, \Delta, t') = \big(1 - f(t')\big)G(\Gamma, \Delta, t') \tag{5}$$

The fraction $f$ has an age dependence with $f(t' < t_i') = 0$, $f(t' > t_f') = 1$ and an exponential form between $t_i'$ and $t_f'$, which are the transition ages between the latitudinally varying and purely tropical surface source regions. Based on Ray et al. (2022) where a value of $t_f' \approx 150$ days was found to be optimal above 380K, we use values of $t_i' = 0.4$ years and $t_f' = 0.6$ years (Fig. S2). The Ray et al. (2022) study showed the importance of the extratropical surface as a source region for certain locations and seasons in the lower stratosphere, such as over North America

during the monsoon season. But for other locations and seasons the tropical surface may be the most important source region for the youngest ages in the stratosphere and the calculation allows for that possibility.

### 3.1 Convolutions



Convolutions of the age spectra with the surface mixing ratio time series for each trace gas are performed for the

time period of available *in situ* measurements. The convolved mixing ratios, $\chi_i$, are given by

$$\chi_i(\Gamma, \Delta, y, t) = \chi_{iTR}(\Gamma, \Delta, t) + \chi_{iV}(\Gamma, \Delta, y, t) \tag{6}$$

where $y$ is a surface source latitude parameter and

$$\chi_{iTR}(\Gamma, \Delta, t) = \int_0^\infty \chi_{ioTR}(t - t', y_{oTR}) g_{TR}(\Gamma, \Delta, t') dt' \tag{7}$$

$$\chi_{iV}(\Gamma, \Delta, y, t) = \int_0^\infty \chi_{ioV}(t - t', y_{ov}) g_V(\Gamma, \Delta, t') dt' \tag{8}$$


The surface mixing ratio time series $\chi_{ioTR}$ are tropical averages from 30°S-30°N in the case of $\chi_{iTR}$, or latitudinally

varying averages in 10° intervals ranging from 60°S-60°N ($y_{ov}$) in the case of $\chi_{iV}$.

Examples of $\chi_{SF6}$ and $\chi_{CO2}$ as a function of $\Gamma$ for early November 1994 are shown in Fig. 2. The dependence of $\chi_i$

on $y_{ov}$ and $R$ are shown separately to reveal the $\Gamma$ range most sensitive to these parameters. $\chi_i$ varies most strongly

with $y_{ov}$ for $\Gamma < 2$ years (Fig. 2a,b). This is expected due to the functional form of $g_V$ which peaks for $t' < 0.5$

years. Even for $\Gamma = 2$ years, $g_V$ is a significant portion of the total age spectrum $G$ (Fig. S2). But for $\Gamma > 2$ years

the contribution of $g_V$ to $G$ decreases rapidly and thus the dependence on $y_{ov}$ as well. The values of $t_i'$ and $t_f'$ are of

course not known precisely and likely vary with time and location as shown for a limited time and region in Ray et

al. (2022). Changes in the assumed time scales of transport from $y_{ov}$ will change the $\chi_i$ distributions in Fig. 2

somewhat but we do not expect moderate time scale changes to change our results. One reason for this is the

frequent opposite dependence of the two trace gas convolutions on $y_{ov}$ so that even if the $\chi_i$ distributions were

broadened based on different assumed transport time scales, the optimal solutions will be confined to the relatively

narrow overlap region of the distributions. We do not explore the effects of the values of $t_i'$ and $t_f'$ any further in this

work but it is an interesting aspect that would be helpful to understand better.

For NH Fall, the contrast between the $\chi_{SF6}$ and $\chi_{CO2}$ dependence on $y_{ov}$ for $\Gamma < 2$ years is stark, with larger values

of $\chi_{SF6}$ for more northern values of $y_{ov}$ and the opposite for $\chi_{CO2}$. This is due to the opposite latitudinal surface

gradients of these trace gases in the preceding NH summer, shown by the minimum of $\chi_{CO2}$ for the most northern

values of $y_{ov}$ at $\Gamma = 0.3$ years. The reversed latitudinal dependence of $\chi_{SF6}$ and $\chi_{CO2}$ persists to $\Gamma \approx 2$ years since

$g_V$ remains influenced by the NH summer for all values of $\Gamma$. Values of $\chi_{CO2}$ for April have the opposite latitudinal

dependence (Fig. S3) as in November due to the opposite latitudinal surface gradient of $CO_2$ in NH winter compared

to summer. For $\Gamma > 3$ years, the values of $\chi_i$ have negligible dependence on $y_{ov}$ and correspond to those using only





$\chi_{ioTR}$ (dashed green lines in Fig. 2). Thus, for older mean ages the values of $\chi_i$ are similar to previous studies that

used the average of Mauna Loa and Samoa station data (e.g., Andrews et al., 2001).

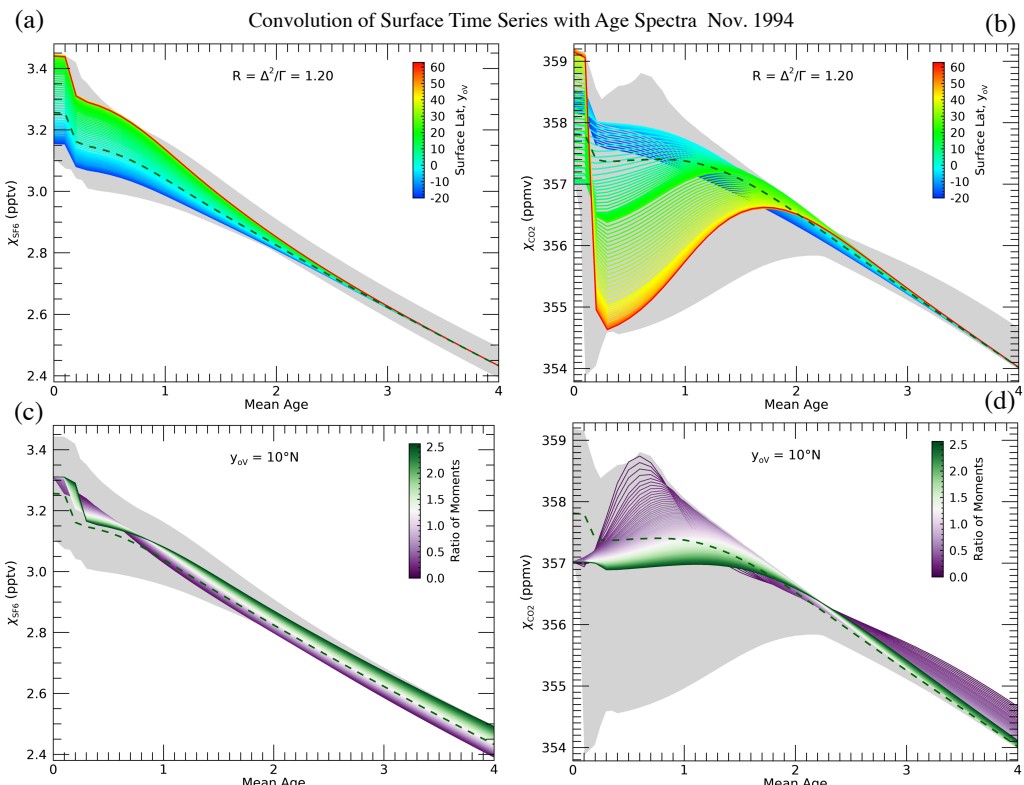

*Figure 2. Convolutions of SF₆ (a,c) and CO₂ (b,d) surface mixing ratios with a range of age spectra and mean ages.*
*These convolutions could be compared to stratospheric measurements of each trace gas in early November 1994*

*(NH Fall). The dependence of the convolutions on surface source latitude is shown in the top row while the*
*dependence on the ratio of moments is shown in the bottom row. The grey shading represents the full range of*
*possible mixing ratios with both source latitude and ratio of moments allowed to vary. The dashed green line*
*represents convolutions with only a tropical surface source.*


The dependence of $\chi_{SF6}$ and $\chi_{CO2}$ on $R$ for $y_{oV}$ =10°N contains a significant spread for nearly all values of $\Gamma$ but

especially for $\Gamma > 2$ years, in contrast to the dependence on $y_{oV}$ (Fig. 2c,d). In NH Fall, $\chi_{SF6}$ and $\chi_{CO2}$ have the

opposite dependence on $R$ for $\Gamma > 1$ year, while for NH Spring the dependence is the same for both trace gases (Fig.

S3c,d). This again reflects the seasonal change of the CO₂ surface gradient and how the shape of the age spectra can

make the convolutions sensitive to a particular season.



A noticeable feature of the NH Fall convolutions is the wide range of possible values of $\chi_i$ for the same $\Gamma$, especially $\chi_{CO2}$ for $\Gamma < 2$ years as shown by the grey shading (Fig. 2). The implication of this is that unless there is high confidence in $y_{ov}$ and $R$ for a particular airmass, the uncertainty in $\Gamma$ from either $SF_6$ or $CO_2$ alone can be

significant. But in combination, the intersection of $\chi_{SF6}$ and $\chi_{CO2}$ can provide a much narrower range of possible $\Gamma$, which we show below to not only reduce the uncertainty in $\Gamma$ compared to using a single trace gas but also reveals optimal values of $y_{ov}$ and $R$.

### 3.2 Adjustments for photochemistry


Before two trace gases can be used in an optimization of age of air, their photochemical lifetimes (Eq. 1) and thus potential adjustments to the measured mixing ratios ($\chi_i^*$), must be considered. A reduction of $\chi_{CO2}^*$ has traditionally been made before $\Gamma$ is calculated due to the conversion of $CH_4$ to $CO_2$ in the atmosphere (e.g., Boering et al., 1996; Daniel et al., 1996). This adjustment requires simultaneous measurements of $CH_4$ or $N_2O$ along with $CO_2$ and has

assumed that the difference between the surface average $CH_4$ at the time of the measurement and the $CH_4$ measured in the stratosphere is entirely due to the oxidation of $CH_4$ to $CO_2$. In the case where a simultaneous $\chi_{CH4}^*$ was not available and $\chi_{N2O}^*$ was, then a $CH_4$-$N_2O$ relationship has been used to approximate $\chi_{CH4}^*$ (e.g., Andrews et al., 2001).

While the adjustment of $\chi_{CO2}^*$ due to $CH_4$ oxidation is typically small since $CH_4$ surface mole fractions have been ~1.7-1.9 ppm over the past few decades, the effect on $\Gamma$ estimates can be significant based on the nonlinear shape of the $\chi_{CO2}$ distributions shown in Figs. 2b and 2d. Furthermore, it has been shown in modeling studies that not all of the oxidized $CH_4$ is converted into $CO_2$ (Boucher et al, 2009; Shindell et al., 2017). Although the average conversion percentage in these two studies was 61% and 88%, essentially all of the conversion in the stratosphere

ends up as $CO_2$. Based on these results we use a 95% average conversion of $CH_4$ into $CO_2$ and we also increase the uncertainty on $\chi_{CO2}^*$ by 5% of the $CH_4$ loss to account for some uncertainty in the conversion percentage.

Another aspect of the $CO_2$ adjustment due to $CH_4$ oxidation that we reevaluate is the use of the surface average $CH_4$ mixing ratio at the time of measurement to subtract from $\chi_{CH4}^*$. The mixing ratios that should be used are the

convolution values $\chi_{CH4}$ since these represent the actual boundary conditions of $CH_4$. The values of $\chi_{CH4}$ have a larger dependence on $y_{ov}$ (Fig.S4) than either the $\Gamma$ or $R$. The difference between the surface average $CH_4$ at the time of stratospheric measurement used in previous studies, and values of $\chi_{CH4}$ is typically 0.02-0.03 ppm but can be greater than 0.05 ppm for $\Gamma < 1$ year. On its own, this difference and the subsequent change in the adjustment of $\chi_{CO2}^*$ is small, but as mentioned earlier can have a nonlinear effect on the age calculation. The use of $\chi_{CH4}$ is also

simply the correct technique to estimate the entry or initial $CH_4$ mixing ratios for any measurement location, and especially in an optimization with multiple tracers.



For SF$_6$ it has been known for some time that mesospheric loss has an impact on age of air calculations with this trace gas (Hall and Waugh, 1998; Hall et al., 1999; Andrews et al., 2001). The old age bias in $\Gamma$ from SF$_6$ was

thought to be largely confined to airmasses in or near the winter polar vortex (e.g., Ray et al., 2017) and not in the mid-latitudes (Engel et al., 2009; Ray et al., 2014). Based on recent studies we now understand that the old age bias in $\Gamma$ from SF$_6$ has a latitudinal and temporal dependence on the mixing ratio of SF$_6$ and that a $\Gamma$ bias correction can be found and applied (Loeffel et al., 2022; Garny et al., 2024). This is an important advancement in the use of SF$_6$ as an age of air tracer and without it an optimization of the kind described here would not be possible.


We make SF$_6$ adjustments based on a correction technique from the work of Garny et al. (2024) that parameterizes the expected $\Gamma$ bias as a function of $\Gamma$ and time. We convert this $\Gamma$ bias into a mixing ratio adjustment via the SF$_6$ tropospheric growth rate over the previous year and add this estimated mesospheric loss to $\chi^*_{SF6}$. There are uncertainties inherent to this technique, primarily due to the relatively new $\Gamma$ bias estimates that are based on a

single model thus far. We do not explicitly add an additional uncertainty to the adjusted values of $\chi^*_{SF6}$ (denoted as $\chi^*_{aSF6}$), but we do investigate possible uncertainties in the adjustments through a scaling method that is described below.

### 3.3 Optimization


We now look at an example of a specific measurement time and place to demonstrate the optimization technique. The example is from an ER-2 flight in early November 1994 during the ASHOE-MAESA campaign at a time when simultaneous $\chi^*_{SF6}$ and $\chi^*_{CO2}$ were available. The range of $\chi_{CO2}$ and $\chi_{SF6}$ values are the same as shown by the grey shading in Fig. 2 since those values are relevant for the November 1994 time period. The photochemically adjusted

$\chi^*_i$ values, denoted as $\chi^*_{ai}(\Gamma)$, have a dependence on $\Gamma$ since as described above, for SF$_6$ the older the $\Gamma$ is the more loss will have occurred. Likewise for CO$_2$ the $\chi_{CH4}$ values are dependent on $\Gamma$. The difference between the $\chi^*_i$ and $\chi^*_{ai}(\Gamma)$ values for both trace gases are shown in Figs. S5 and 3. The uncertainty on $\chi^*_{ai}(\Gamma)$ is indicated by $\chi^*_{eai}$ but for the two species used here only CO$_2$ has a change in the size of the uncertainty from the original measurement due to the uncertainty in CH$_4$ production.


A range of possible mean ages, surface source latitudes and ratios of moments ($\Gamma$, $y_{ov}$ and $R$) can be found for each trace gas based on agreement of the convolutions ($\chi_i$) with the adjusted measured mole fractions ($\chi^*_{ai}(\Gamma)$) within the uncertainty range ($\pm\chi^*_{eai}$). For the example shown here, the values of $\Gamma$ compatible with $\chi^*_{aCO2}(\Gamma)$ range from 0.1-2.6 years, while for SF$_6$ the $\Gamma$ range is 2.4-2.9 years (Figs. S5 and S6). Figure 3 zooms in on the range of $\Gamma$ and $\chi_i$

around where both trace gases have similar $\Gamma$ solutions. The $\chi_{SF6}$ values that agree with $\chi^*_{aSF6}(\Gamma)$ within $\pm\chi^*_{eSF6}$ are shown by the magenta symbols and labeled $\chi_{SF_6}(\Gamma_{SF6})$ in Fig. 3a and likewise for CO$_2$ in the orange symbols and labeled $\chi_{CO_2}(\Gamma_{CO2})$ in Fig. 3b. The $\Gamma$ resolution of our calculation was 0.05 years which is why there are distinct columns of symbols in the figure.




Since any optimal solutions must be compatible with both trace gases, we need to look at how the possible $\Gamma, \Delta, y$ combinations from one tracer affect the $\chi_i$ values of the other tracer. The values of $\chi_{SF_6}$ with $CO_2$ solutions are shown by the orange symbols in Fig. 3a and labeled $\chi_{SF_6}(\Gamma_{CO2})$ and likewise the values of $\chi_{CO_2}$ with $SF_6$ solutions are shown by the magenta symbols in Fig. 3b and labeled $\chi_{CO_2}(\Gamma_{SF6})$. Most of the $\chi_{SF_6}(\Gamma_{CO2})$ values are greater than the $\chi_{aSF6}^*$ values within $\pm\chi_{eSF6}^*$, while most of the $\chi_{CO_2}(\Gamma_{SF6})$ values are less than the $\chi_{aCO2}^*(\Gamma)$ values within

$\pm\chi_{eaCO2}^*(\Gamma)$. However, there is a region of overlapping solutions for both trace gases as indicated by the green symbols and labeled $\chi_{SF6}(\Gamma_\cap)$ and $\chi_{CO2}(\Gamma_\cap)$ where $\Gamma_\cap$ represents the intersection of possible $\Gamma, \Delta, y$ combinations.

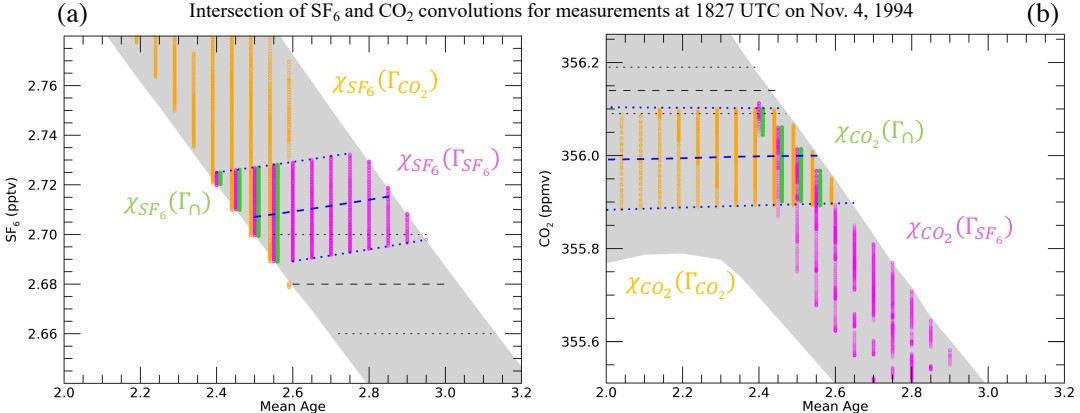

Figure 3. *Zoomed in convolutions similar to Fig. 2 but also including the simultaneous measured SF₆ and CO₂*
*mixing ratios from the ASHOE-MAESA ER-2 flight on Nov. 4, 1994 at 1827 UTC in the black dashed lines with uncertainties indicated by the dotted black lines. The adjusted mixing ratios and uncertainties are indicated by the dashed and dotted blue lines. The range of solutions for SF₆ alone ($\chi_{SF_6}(\Gamma_{SF_6})$) are indicated by the magenta symbols in (a) and likewise for CO₂ alone ($\chi_{CO_2}(\Gamma_{CO2})$) by the orange symbols in (b). The values of $\chi_{SF_6}$ with the CO₂ solutions ($\chi_{SF_6}(\Gamma_{CO2})$) are shown by the orange symbols in (a) and likewise the values of $\chi_{CO_2}$ with the SF₆*
*solutions ($\chi_{CO_2}(\Gamma_{SF6})$) are shown by the magenta symbols in (b). The intersection of solutions that agree with both trace gases within uncertainties are indicated by the green symbols and labeled $\chi_{SF6}(\Gamma_\cap)$ and $\chi_{CO2}(\Gamma_\cap)$.*

The intersection of solutions is the basis of the optimization used to find a single best solution for each set of measurements. We first find normalized differences expressed as $\delta\chi_i = |\chi_i - \chi_{ai}^*|/\chi_{eai}^*$ (Fig. S6). A combined
difference quantity, $\delta\chi_\cap$, for the intersection of solutions is defined as the average of the $\delta\chi_i$ values (Fig. S7),

$$\delta\chi_\cap = (\delta\chi_{CO_2}(\Gamma_\cap) + \delta\chi_{SF6}(\Gamma_\cap))/2. \tag{9}$$

From $\delta\chi_\cap$ we define weighting functions $W_\cap = (\max(\delta\chi_\cap) - \delta\chi_\cap)/\sum \delta\chi_\cap$ to be multiplied by the set of transport
parameters within the intersection of solutions to find the optimum values





$$\Gamma_o = W_\cap \Gamma_\cap, \ \Delta_o = W_\cap \Delta_\cap, \ y_o = W_\cap y_\cap. \tag{10}$$

Thus, for each measurement location there is a single set of transport parameters $\Gamma_o, \Delta_o, y_o$ that are optimized for $\chi_{ai}^*$

of both trace gases (Fig. S8).

Note that the optimization can be performed for a single trace gas by following the same general procedure
described above. The range of possible transport parameters is larger for a single trace gas compared to an
optimization with two or more trace gases and so the uncertainty on the results will be larger, and there will be a

higher probability that the optimized transport parameters are 'incorrect'. In the example measurement location
used here, if the optimization were performed only on $CO_2$, $\Gamma_o$ would be much younger than the result with both $SF_6$
and $CO_2$.

### 3.4 Offsets to measured mixing ratios


In some cases, the intersection of possible solutions for $SF_6$ and $CO_2$ within uncertainties is small or there is no
intersection at all. This lack of overlap could be due to various reasons, such as an inconsistent calibration between
surface and atmospheric measurements or an inaccurate correction for photochemical loss or production of a trace
gas. Since there can be only one set of transport parameters in the real atmosphere at any one time and place, a lack

of an intersection of solutions implies then an offset in one or both trace gases at the measurement location is
necessary to perform the optimization.

To account for the possibility that an offset of either trace gas is needed to obtain a solution set, we ran an ensemble
of optimizations (see Supplement) that swept over a grid of mixing ratios with positive and negative offsets from the

measured value of each trace gas for each measurement location. Note that we offset the adjusted ($\chi_{ai}^*$) mixing
ratios and use the same uncertainties ($\chi_{eai}^*$). The optimization performed for each member of the offset ensemble
produces a unique set of transport parameters $\Gamma_o(s), \Delta_o(s), y_o(s)$ and a minimum value of the combined normalized
difference $\delta\chi_{\cap min}(s)$, with the dependence on $s$ added to denote the offset ensemble member (Fig. S9).

There are a number of possible ways to select the best member of the ensemble and thus the best overall set of
solutions. We chose to use an optimization method that is consistent with that used for each member of the
ensemble as described above and the details are included in the supplement. Most of the offset values are small,
especially for $CO_2$, but there are certain measurement times and locations where the optimal offset of $SF_6$ is
significant. Figs. 4 and 5 show $SF_6$ and $CO_2$ average optimal offset values as a function of normalized $N_2O$ for the

1990s and 2020s data. Normalized $N_2O$ ($\chi_{N2On}$) is defined as $\chi_{N2O}^*(t)/\overline{\chi_{N2Oo}}(t)$, that is, the measured atmospheric
$N_2O$ mixing ratio divided by the global mean surface $N_2O$ mixing ratio at the time of measurement. $N_2O$ has often
been used as a quasi-vertical coordinate when displaying *in situ* data due to the compact relationships of long-lived



trace gases and $\Gamma$ with $N_2O$ (e.g. Plumb and Ko, 1992; Andrews et al., 2001). We use normalized $N_2O$ in this figure and others in order to account for the growth rate of $N_2O$ over time, enabling the comparison of compact

relationships with $N_2O$ over years or decades.

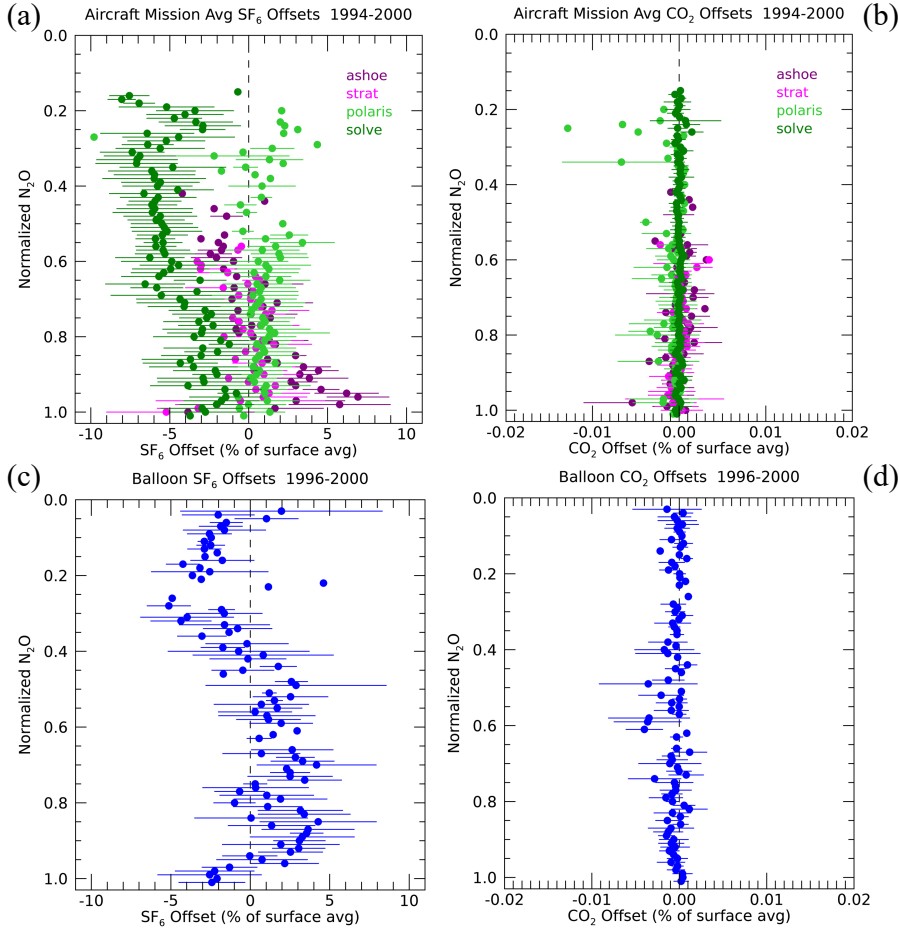

Figure 4. *Offset values ($\chi_{is}$) divided by the global average surface mixing ratios of each trace gas at the time of*

*measurement and binned by normalized $N_2O$ ($\chi_{N2On}$) for $SF_6$ (a,c) and $CO_2$ (b,d). The top row (a,b) shows*
*normalized offset values binned separately for four different aircraft missions in the 1990s and the bottom row (c,d)*
*shows normalized offset values averaged over seven OMS balloon flights in the 1990s.*

The offset values are also normalized by the global average surface mixing ratios of each trace gas and displayed as

a percentage. There are several features of note in Figs. 4 and 5. The first is that $\chi_{CO2s}$ is essentially zero, <0.01%
compared to an annual growth rate of 0.3-0.5%, for all values of $N_2O$ for both the 1990s and 2020s data. This



follows from the symmetric nature of the offset ensemble results in $CO_2$ as shown by the example in Fig. S9. This implies that there is no advantage in the optimization calculation to offset $CO_2$ either positively or negatively in order to better agree with any set of $SF_6$ solutions. Further implications are that the calibration between $CO_2$ measured *in situ* in the atmosphere and the surface have remained consistent over several decades and that the photochemical adjustment of $CO_2$ does not appear to have a significant bias.


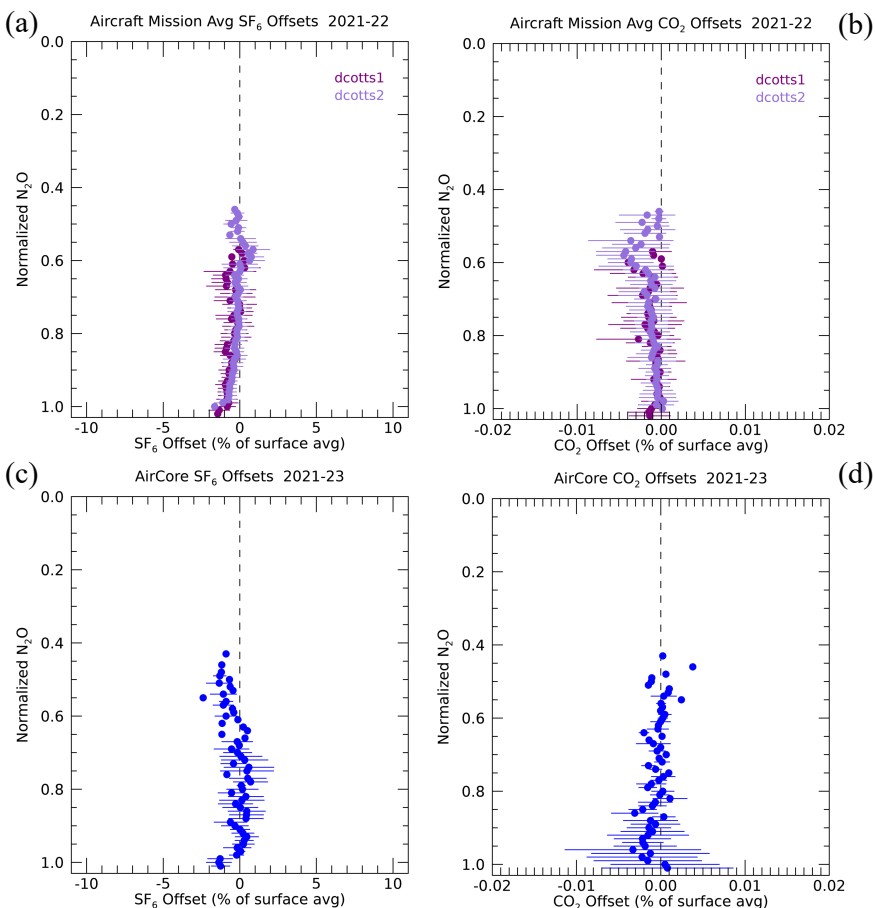

Figure 5. *Same as Fig. 4 but for two DCOTSS aircraft missions in the 2020s (a,b) and for routine AirCore balloon flights in the 2020s (c,d).*


A second feature of note is the size of $\chi_{SF6s}$ for the 1990s data compared to that from the 2020s. In the 1990s, $\chi_{SF6s}$ ranges from -9% to +7% at a time when the annual growth rate of $SF_6$ was ~5%/year, while in the 2020s $\chi_{SF6s}$ ranges from -2% to +1% at a time when the annual growth rate was ~3.5%/year. Thus, the $\chi_{SF6s}$ values translate to $\Gamma$ shifts from -1.8 to +1.4 years in the 1990s and -0.5 to +0.3 years in the 2020s. As mentioned earlier, these $\chi_{SF6s}$ values could be due to an inaccurate $SF_6$ photochemical loss adjustment that we apply in the optimization based on






the Garny et al. (2024) study. If the photochemical loss was estimated to be larger than reality, $\chi^*_{aSF6}$ would be too large and the optimization would compensate by producing negative $\chi_{SF6s}$ values to agree with $CO_2$ and vice versa. The fact that there is essentially an equal amount of positive and negative $\chi_{SF6s}$ values, outside of the SOLVE mission that we will discuss below, and that the 2020s $\chi_{SF6s}$ values are mostly near zero suggest that there is not a systematic error in the photochemical loss adjustments. There can be certain locations and times when the adjustment could be too large or small but this is likely due to known limitations of the simplified adjustment used here, such as in the region of the polar vortex where the gradients of $SF_6$ loss are large.

A third feature, notable in Fig. 4, is the nearly linear dependence and large negative values of $\chi_{SF6s}$ as a function of $N_2O$ during the SOLVE mission. These large negative values could be due to either assumed photochemical loss that is much too large or a calibration issue with $\chi^*_{SF6}$ during the mission. Based on the outlier nature of $\chi_{SF6s}$ during SOLVE compared to other missions as well as the size of the photochemical loss error that would be necessary to cause these values (>100% for $\chi_{N2On}$>0.3) it is unlikely the assumed photochemical loss is the primary cause. There was a known $SF_6$ calibration scale change during SOLVE (https://espoarchive.nasa.gov/archive/browse/oms/Balloon) that resulted in the value of the Volk et al. (1997) delay term from the surface to the tropical tropopause ($\delta\Gamma$) to be changed from +0.8 years used in previous missions to -0.8 years in SOLVE. A negative value of $\delta\Gamma$ is obviously non-physical but was necessary to calculate reasonable $\Gamma$ values due to the relatively large $\chi^*_{SF6}$ compared to the surface values. This change in the value of $\delta\Gamma$, even considering the different source region of the surface compared to the tropical tropopause, is enough to explain most of the negative $\chi_{SF6s}$ values from the optimization.

### 3.5 Single Trace Gas Optimizations

In order to utilize the large number of *in situ* measurements with no simultaneous or near-simultaneous measurements of a second age of air trace gas, we also perform the optimization described here on a single trace gas. A single trace gas optimization is not ideal but we make use of the relatively tight $\Gamma - \chi_{N2On}$ and $R - \chi_{N2On}$ correlations derived from the two trace gas optimizations to constrain those parameters for a single trace gas optimization (Figs. S10 and 6). Based on the measured $\chi_{N2On}$ at any location we choose a range of possible $\Gamma$ and $R$ consistent with the correlations based on other missions during a similar time period and perform the optimization only within the limited range of each parameter. Thus, our technique requires a near-simultaneous measurement of $N_2O$ with any single age of air trace gas.

The lack of simultaneous measurements can be due to a mismatch in instrument sampling rates, as is the case for $CO_2$ and $SF_6$ mentioned in Sect. 2, or the lack of measurements of two different age of air trace gases on a particular airborne platform or mission. The single trace gas optimization is often most desirable with $CO_2$ since there are usually nearly two orders of magnitude more measurements available during a mission compared to $SF_6$. Since $\chi_{CO2s}$ is negligible for all of the measurements considered in this study, we can reasonably perform the optimization



with $CO_2$ alone without scaling. In the 2020s, $\chi_{SF6s}$ is also minimal so we can perform the single trace gas

optimization with $SF_6$, for instance, with data from the SABRE mission in 2023 when $CO_2$ measurements were not

available.

**4 Results**

*4.1 1990s*

The 1990s was the last time period of extensive *in situ* measurements from both aircraft and balloon platforms of

long-lived trace gases and $\Gamma$ calculated from $CO_2$ and $SF_6$ (denoted $\Gamma_{CO2}$ and $\Gamma_{SF6}$) in the stratosphere at altitudes

above 16 km. The $\Gamma_{CO2}$ and $\Gamma_{SF6}$ values from this time period have been used extensively over the past several

decades to compare to model output, especially the use of the 'wing plot' of $\Gamma$ in the 18-20km altitude range as a

function of latitude (Waugh et al., 1997; Hall et al., 1999; Andrews et al., 2001; Waugh and Hall, 2002; Strahan et

al., 2011; Diallo et al., 2012; Chabrillat et al., 2018; Dietmuller et al., 2018; Ploeger et al., 2019). The compact $\Gamma -$

$\chi_{N2On}$ relationship has also been commonly used as a transport metric and method to compare *in situ* data and model

output (e.g. Hall et al., 1999; Andrews et al., 2001; Strahan et al., 2011; Birner et al., 2020). As mentioned in Sect.

3.5, we utilize the $\Gamma_o - \chi_{N2On}$ relationship in our optimizations for cases with a single trace gas measurement and

we also can compare the relationships found here with those from previous $\Gamma$ techniques.

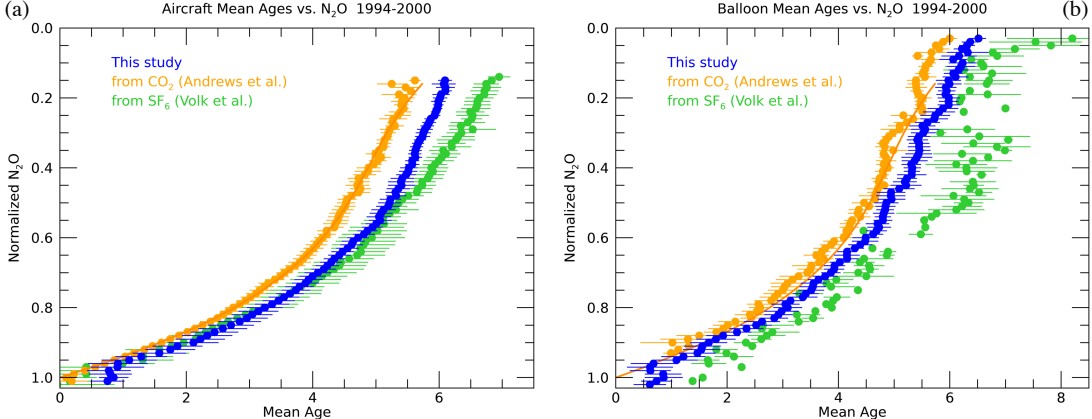

Figure 6. *Mean age vs. normalized $N_2O$ for aircraft (a) and balloon (b) measurements from the 1990s. The results*

*from this study are shown in blue, those based on previously published $CO_2$ in orange and from $SF_6$ in green. The*

*aircraft measurements are from the four ER-2 missions shown in Fig. 4a. The uncertainties represent the standard*

*deviation of the mean ages within each $N_2O$ bin. The functional form from Andrews et al. (2001) translated to*

*normalized $N_2O$ is shown as the solid orange line in both plots.*



Figure 6 shows $\Gamma - \chi_{N2On}$ correlations for all of the 1990s aircraft and balloon data. The $\Gamma_{SF6}$ bias of up to two years due to $SF_6$ photochemical loss at that time is seen by comparing the $\Gamma_{SF6}$ and $\Gamma_{CO2}$. Values of $\Gamma_o$ typically fall between the values of $\Gamma_{SF6}$ and $\Gamma_{CO2}$. We clearly expect $\Gamma_o$ to be younger than most of the $\Gamma_{SF6}$ values since we account for photochemical loss, while we also expect $\Gamma_o$ to be older than $\Gamma_{CO2}$ by at least three months since the source region of our calculation is the surface compared to the tropical tropopause for the previous calculation.


The aircraft $\Gamma_{CO2}$ from the 1990s are based on $\chi_{N2O}^*$ rather than $\chi_{aCO2}^*$ for values of $\chi_{N2On} > ~0.8$ (Andrews et al., 2001), which explains the tight relationship and no variability in $\Gamma_{CO2}$ in this range in Fig. 6a. This relationship was based on measurements in the tropical tropopause region during this time. While this $\Gamma_{CO2} - \chi_{N2On}$ relationship was valid for the particular time and place of the original measurements, a fixed relationship of this sort does not allow

detection of changes in the relationship over time or allow for variation in latitude. The $\Gamma_o$ values in this study are derived independently of any $\Gamma - \chi_{N2On}$ relationship and yet produce very compact curves for both the aircraft and balloon data as seen in Fig. 6. There are at least an order of magnitude more aircraft measurements compared to balloon measurements resulting in the relatively more compact relationships in Fig. 6a compared to those in Fig. 6b.

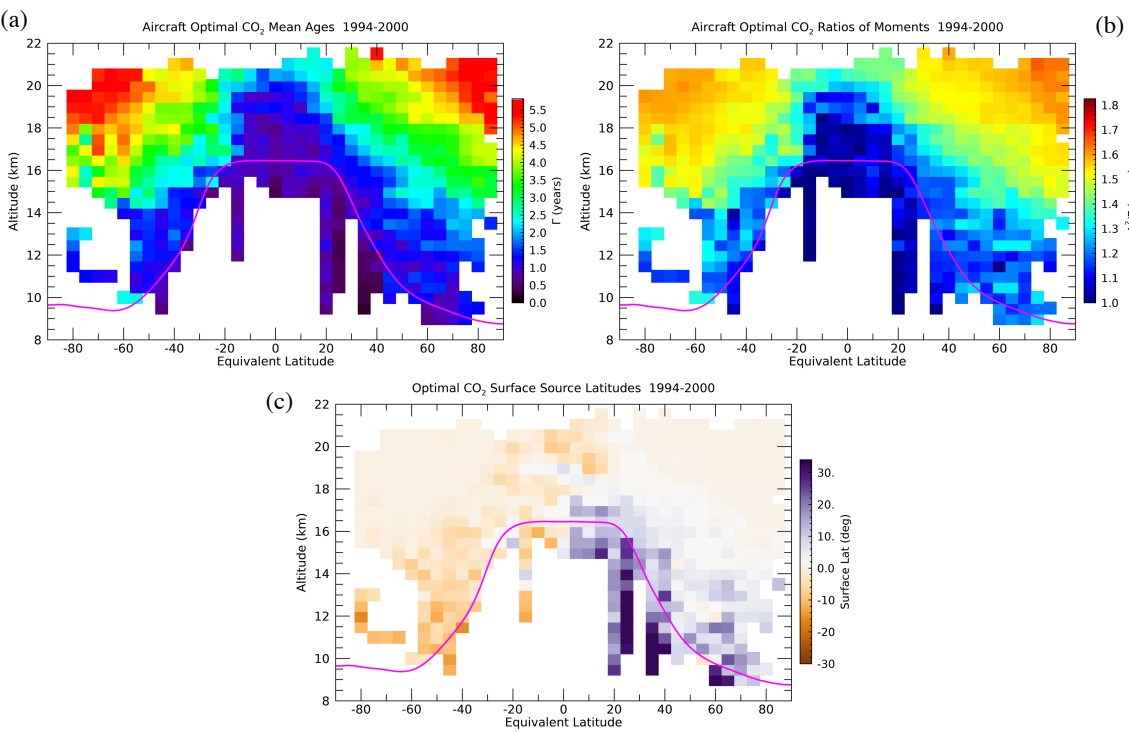


Figure 7. *Equivalent latitude vs. altitude distributions of mean ages (a), age spectra ratios of moments (b) and surface source latitudes (c) from the $CO_2$ optimizations averaged over the 1990s aircraft missions.*



Average distributions of the transport parameters derived from the optimizations with the 1990s aircraft
       measurements are shown in Figs. 7 and S11. The two figures reveal the difference in spatial coverage of where there
       were near simultaneous measurements of $SF_6$ and $CO_2$ (Fig. S11) compared to $CO_2$ alone (Fig. 7). We focus here on
       the $CO_2$ optimizations shown in Fig. 7 since the more extensive number of measurements and spatial coverage gives
       a nearly complete picture of the average transport features in the upper troposphere and lower stratosphere.


       One of the most notable features of the distributions in Fig. 7 is that they cover a range of latitudes and altitudes
       including the troposphere, lowermost stratosphere and stratospheric overworld nearly from pole to pole. There have
       been no previous $\Gamma$ estimates from *in situ* measurements that span this spatial range. This is partly due to the
       original concept of age of air as a time scale since an air parcel crossed the tropical tropopause into the stratospheric

overworld (Kida, 1983). Thus, many previous $\Gamma$ estimates from *in situ* data have focused only on the stratospheric
       overworld with the tropical tropopause as the entry region (e.g. Boering et al., 1996; Volk et al, 1997; Andrews et al.,
       2001; Engel et al., 2008). Subsequent studies have focused on the lowermost stratosphere with a modified age of air
       concept to account for multiple tropopause entry regions (e.g. Boenisch et al., 2009; Hauck et al., 2019, 2020). And
       interhemispheric tropospheric age of air from the NH surface to the SH surface has also been estimated from *in situ*

surface measurements (Waugh et al., 2013; Holzer and Waugh, 2015). The Ray et al. (2022) study calculated $\Gamma$
       from the troposphere through the stratospheric overworld but just from one aircraft mission in the NH midlatitudes.

       In general, the $\Gamma_o$ distributions look as expected with the youngest ages in the troposphere, near the tropopause as
       well as in the tropical stratosphere, and the oldest ages at the highest latitudes and altitudes. Values of $\Gamma_o$ at the

tropopause vary with latitude, from ~0.5 years in the tropics to 1-1.5 years in the extratropics. The extensive *in situ*
       measurements of key long-lived trace gases in the 1990s provides a nearly complete view of the $\Gamma_o$ distributions at
       that time.

       The distribution of $R_o$ shown in Fig. 7b is a unique contribution of this study. The $R_o$ distribution mirrors that of $\Gamma_o$

with the smallest values near one year in the troposphere and tropical stratosphere, increasing to values >1.5 years at
       high latitudes and altitudes. The only other previously published distributions of $R$ come from model output. Hall
       and Plumb (1994) found values from 0.4-0.7 years throughout the stratosphere from global climate model output at
       that time. Based on this study, Engel et al. (2008) used a value of $R = 0.7\pm0.5$ years in their analysis of $\Gamma$ trends
       from balloon measurements of $CO_2$ and $SF_6$. In contrast, Volk et al. (1997) used a value of $R = 1.25\pm0.5$ years to

calculate mean ages from $SF_6$ measurements based on the model output from Waugh et al. (1997). Thus, the value
       of $R$ has varied widely among models and has essentially been unconstrained by observations. More recent
       modeling studies have revealed the sensitivity of $R$ to the long age tail of $G$ (e.g. Ploeger and Birner, 2016) and have
       found values of $R$ up to 1.7 years in the extratropical lower stratosphere with appropriate extension of the age
       spectra tail (Fritsch et al., 2020). The magnitude and distribution of the $R$ values shown in Fig. 7b compare well to



those in Fritsch et al. (2020, Fig. 6b) and provide some of the first extensive validation of modeled values of $R$
throughout the lower stratosphere.

The surface source latitude distribution range in Fig. 7c is also a unique result with Ray et al. (2022) the only known
previous study to have calculated this quantity from trace gas observations, but over a more limited time and

location. The latitudes shown in the figure are averages of $y_{ov}$ scaled by $\sum g_v$, which could be considered an
extratropical fraction of air. We scale $y_{ov}$ this way to show where the extratropical surface source latitudes matter in
the optimization, which is primarily for locations where $\Gamma < \sim 3$ years. For $\Gamma$ older than this the extratropical
surface contribution to the optimization is negligible and the source latitude is the equator. The most notable feature
in the source latitude distribution is the hemispheric symmetry with surface latitudes from the same hemisphere as

the location of $\chi_i^*$ except for in the tropical stratosphere. There will of course be seasonal variability in this quantity
which we will not describe further here. This quantity could also be used to compare to surface source region
attributions in the stratosphere from model output (e.g. Yan et al., 2021).

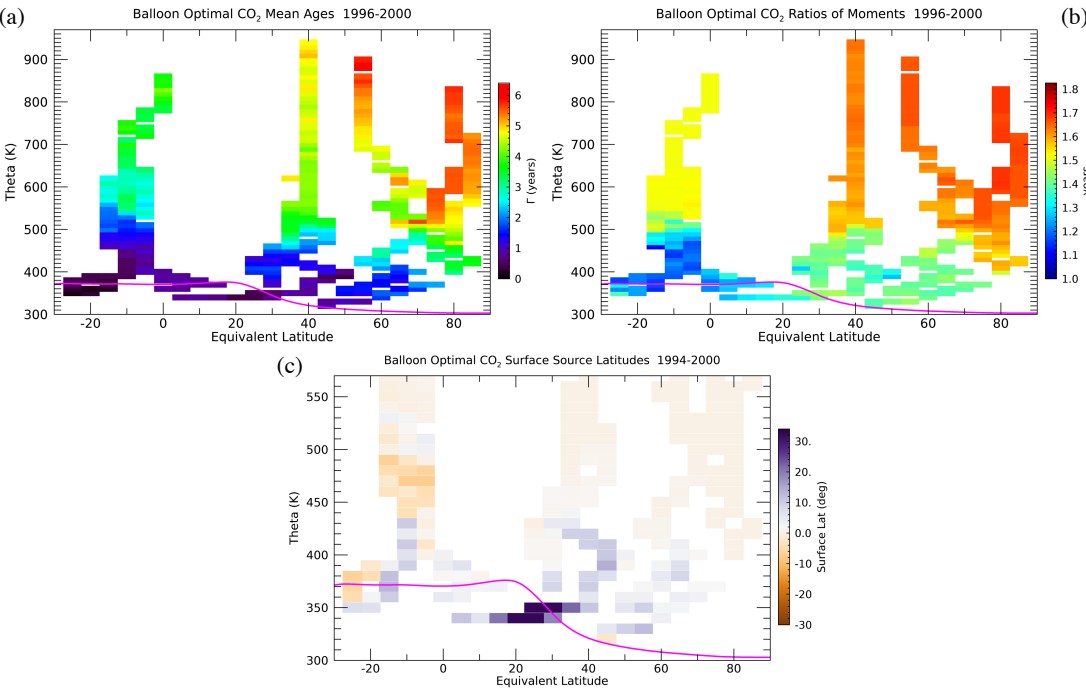

Figure 8. *Similar to Fig. 7 but for OMS balloon $CO_2$ optimizations averaged over the 1990s and the vertical
coordinate is potential temperature. Note that the latitude scale has been changed from Fig. 7 to better match the
balloon sampling and the theta scale is cut off for the surface source latitudes.*



Distributions of the three transport quantities from the optimizations based on OMS balloon measurements in the

1990s (Fig. 8) show the more limited sampling compared to the aircraft, primarily in latitude, but the more extensive

vertical coverage.  The vertical scale on these plots is potential temperature rather than altitude since this emphasizes

the stratospheric overworld more than the tropopause region.  The balloon measurements extend up to 900 K (34 km

altitude), much higher than the ~550 K (22 km) level that can be reached by aircraft.  No other *in situ* platform can

fly that high, which is of particular value in the tropics.


The OMS balloon $\Gamma_o$ and $R_o$ values generally agree with those from the aircraft measurements in similar regions

during this time period.  In the tropics above 600 K, $\Gamma_o$ and $R_o$ are nearly constant at values of ~3 and 1.5 years.  In

the northern extratropics, $\Gamma_o$ reaches 5 years at 900 K and 40°N and 6 years in a deep region of the polar vortex.

Values of $R_o$ peak at ~1.7 years and are nearly constant above 600 K throughout the extratropics.  This is in contrast

to model estimates that have peak values in the extratropical lower stratosphere and decreasing values up to 10 hPa

(Fritsch et al., 2020).  The surface source latitudes have similar features to those from the aircraft measurements in

the NH.  The tropical lower stratosphere has a layered structure of alternating NH and SH influence (Fig. 8c).  There

is some indication of this in the aircraft data as well but the averaging over all seasons shown in Fig. 7c has reduced

the amplitude.


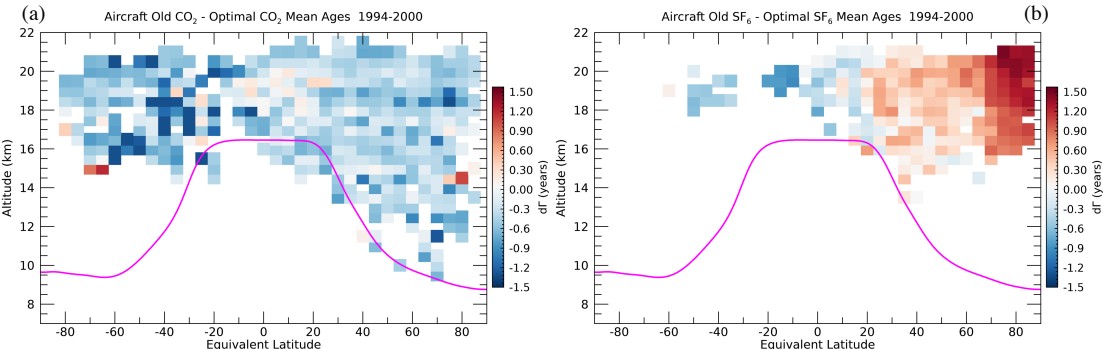

Figure 9. *Gridded average differences between the archived aircraft $CO_2$ (a) and $SF_6$ (b) mean ages minus the*

*optimized mean ages in this study.  The optimized mean ages are from the single trace gas technique with $CO_2$ (a)*

*and $SF_6$ (b) in order to maximize the number of comparison data points for each trace gas.*

Similar to the comparisons in Fig. 6, we can also compare the differences in the spatial distributions of $\Gamma_o$ with $\Gamma_{SF6}$

and $\Gamma_{CO2}$ (Fig. 9).  In the comparisons with $\Gamma_{CO2}$ (Fig. 9a), most locations have a difference of -0.3 to -0.5 years that

represents the transport time between a source region at the tropopause for $\Gamma_{CO2}$ and the surface for $\Gamma_o$.  There are

some locations with larger differences of -1 year, especially in the SH.  The region of these larger age differences in

the SH coincides with where $\Gamma_o$ values are ~1-3 years and the surface source latitudes have SH influence (Fig. 7).



The most negative differences were from measurements taken in October (not shown) which has a similar latitudinal dependence in $\chi_{CO2}$ as in November (Fig. 2b). At this time of year, a SH surface source results in larger $\chi_{CO2}$ values compared to the Mauna Loa-Samoa average and thus a relatively older $\Gamma$ compared to assuming only a tropical source.


The $\Gamma_{SF6}$ values are only younger than $\Gamma_o$ in the tropics and southern midlatitudes (Fig. 9b). The tropics have the youngest mean ages and thus the smallest correction for $SF_6$ so as with $CO_2$, the archived mean ages will be younger in this region due to the tropopause vs. surface boundary conditions. In the northern extratropics the mesospheric loss of $SF_6$ and subsequent old age bias overcomes the 3-6 month negative age bias due to the tropopause versus surface source region difference resulting in positive mean age differences of up to 1.5 years. This distribution of old age bias generally follows that from the model output of Garny et al. (2024) which was the source of the age bias estimates used in our optimization.


### 4.2 2020s


Only in recent years have simultaneous measurements in the stratospheric overworld of the four trace gases utilized in the optimization become available again. This is an unfortunate gap in the monitoring of stratospheric circulation over the past two decades. With the DCOTSS and SABRE aircraft missions as well as AirCore balloon flights, we now have the ability to calculate age of air from *in situ* measurements in the current stratosphere. The new measurements were the inspiration for this study since a consistent technique is necessary to compare the old and new measurements and age of air. Here we briefly describe the derived mean ages, which have not previously been published from this data, and other transport quantities from the three measurement campaigns in the 2020s. A follow up study will examine the differences between the 1990s and 2020s ages in detail



The distributions of $\Gamma_o - \chi_{N2On}$ for the three different mission data sets are compact and agree well on average, with slightly older $\Gamma_o$ from AirCore primarily from the winter sampling (Fig. 10). The summer average AirCore mean ages agree within uncertainties with the DCOTSS mission which took place during the NH summer. The distribution of $\Gamma_o$ in latitude and theta shows the different sampling between the three missions (Fig. 11). DCOTSS used the ER-2 and focused on the summer NH stratosphere which resulted in sampling of relatively young $\Gamma_o$ up to nearly 500 K and at all latitudes. SABRE used the WB-57 and took place in NH winter with a focus on the polar region where relatively old $\Gamma_o$ were sampled, although only $SF_6$ was measured during this mission. AirCore uses balloons flown predominantly from Boulder, CO but in various seasons with the oldest $\Gamma_o$ above 500 K in the midlatitudes. Distributions of $R_o$ and surface source latitudes for each mission are shown in the supplement (Figs. S12, S13).



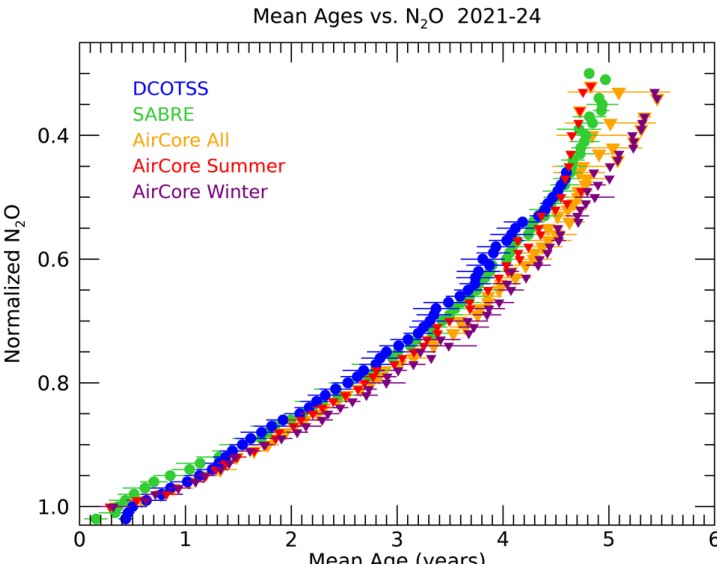

Figure 10. *Distributions of mean age vs. normalized N₂O from the DCOTSS (blue), SABRE (green) and AirCore (orange, red, purple) measurements.*


NOAA's AirCore program is ongoing, and is likely to expand to different latitudes as new recovery techniques for this system are developed, but regardless is a valuable addition to the monitoring of the stratospheric circulation. It is unclear what future aircraft missions will occur with the requisite payload to use this optimization technique and calculate age of air. However, it is unlikely that any period of aircraft sampling of the stratosphere from pole to pole

as in the 1990s will occur again.



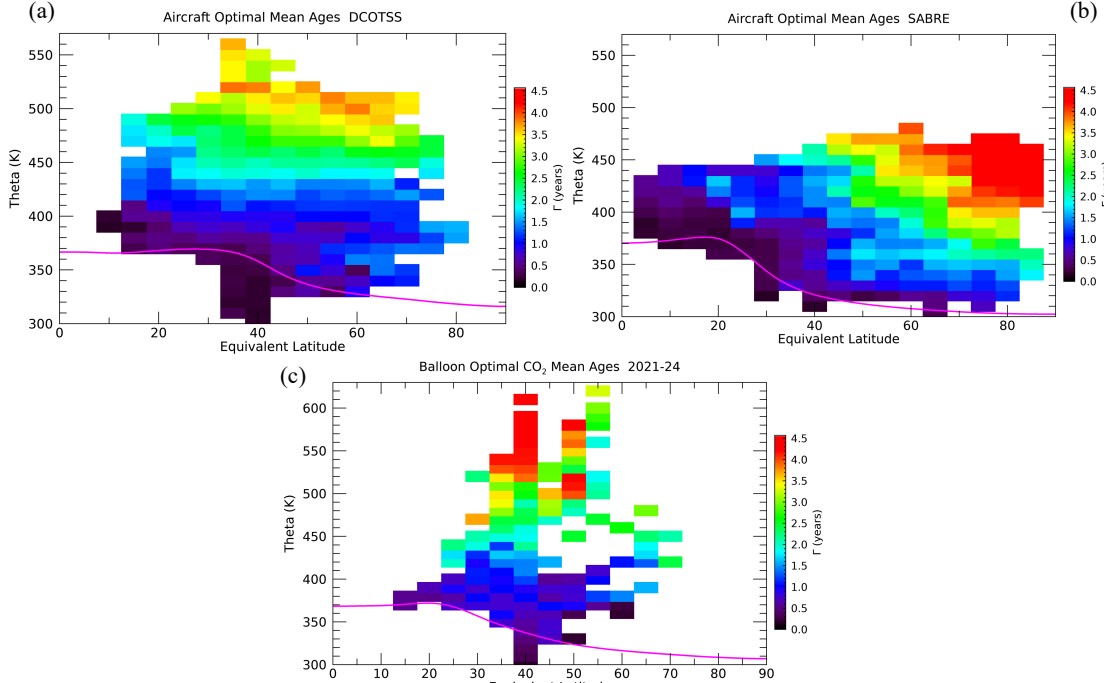

**Figure 11.** *Equivalent latitude vs. potential temperature distributions of mean age from the DCOTSS (a), SABRE (b) and AirCore (c) missions.*


## 5 Summary and Discussion

In this study we describe and demonstrate a new technique to calculate the first and second moments of the age of air distribution as well as surface source latitudes based on aircraft and balloon *in situ* measurements of $SF_6$, $CO_2$,

$CH_4$ and $N_2O$ throughout the upper troposphere, lowermost stratosphere and stratospheric overworld. This technique depends crucially on the recent quantification of the time dependent bias in $SF_6$-derived mean age due to mesospheric loss by Garny et al. (2024) and the NOAA Greenhouse Gas MBL Reference product (Lan et al., 2023) as a time and latitudinally varying surface source for each of the trace gases mentioned above. The main advantages of this new technique are:


1) The use of multiple trace gases allows a single, optimized age of air estimate to be found for each measurement location while minimizing the impact of various uncertainties associated with age of air from single trace gases.

2) Allows the calculation of age of air from the upper troposphere through the stratosphere.

3) Flexibility to utilize measurements from different instruments and platforms across decades and account for calibration offsets.



The technique we use is based on the convolution method of estimating age of air and builds on studies such as Bönisch et al. (2009), Ray et al. (2017), Leedham Elvidge et al. (2018) and Ray et al. (2022). One of the primary aspects of how we define the age of air is that it is from the Earth's surface rather than the tropopause as has been commonly used in previous studies. The main reason we choose to use the Earth's surface is that this is where we have long-term measurements of the trace gases used here, which is necessary for any age of air calculation. Of course, all age of air studies have used surface measurements to generate a source region time series for the trace gas(es) of interest, but for those that used the tropopause as a source region, assumptions about how surface trace gas time series translate to the tropopause need to be made.

There are benefits and complications to the choice of the surface as the age of air source region. Complications include (1) significant latitudinal gradients in most trace gases that are in growth, which makes defining a single source region time series difficult, and (2) conceptualizing what age of air from the surface means compared to the more traditional age since an air parcel passes through the tropopause. We address the first complication by partitioning the surface source region time series of each trace gas into two parts, one averaged over all tropical latitudes similar to the standard Mauna Loa-Samoa average (e.g. Boering et al., 1996; Andrews et al., 2001) and a second part averaged over 10° latitude bins from 60°S-60°N that is a free parameter in the calculation (e.g. Ray et al., 2022). The latitudinally varying surface time series is only considered as a source region for the youngest ages of air, on the order of months, primarily because for older ages the measured trace gases lose sensitivity to the surface latitudinal gradients due to mixing. In other words, unlike an idealized tracer in a model that can be emitted and tagged from a certain surface latitude and followed throughout any trajectory and time scale in the atmosphere (e.g. Orbe et al., 2015), real trace gas measurements can only be identified as having been emitted from a certain surface latitude range for a limited time. Conceptually, this means most of the ages in the age spectra in our calculation are from a tropical average surface source similar to that from almost every previous age of air study but without any assumed constant transport time from the surface to the tropopause. The youngest ages in the age spectra have a flexible surface source latitude that allows us to utilize any latitudinal emission information that persists in the measured trace gases.

Thus, the complications from the use of the surface as a source region can be turned into a benefit in that additional transport information can be obtained from the trace gas measurements. This benefit is only realized with the use of multiple trace gases since any single trace gas can have too many ambiguities in surface sources combined with transport time scales to constrain the latitudinal source region. The more *in situ* trace gas measurements are available, the more potential there is to constrain the surface source region (e.g. Ray et al., 2022). However, most aircraft and balloon *in situ* payloads have not included instruments that measured more than the four age of air trace gases considered here and often only one or two were measured.



Further benefits of a surface source region include consistency with age of air from CCM output and the ability to calculate age of air at any location in the atmosphere. This last benefit applies primarily to locations in the upper
troposphere and above since the shortest transport time scales in the troposphere (days to weeks) require a range of shorter-lived trace gases to define (e.g. Luo et al., 2018). A single technique to calculate age of air in the upper troposphere, tropopause region, lowermost stratosphere and stratospheric overworld can provide new insight into transport in these regions and connect various airborne measurements with different sampling limitations.

A further unique aspect of this study is the offset ensemble as part of the optimization procedure. The assumption behind this technique is that there are many uncertainties in calculating the age of air that are not accounted for in the measurement errors alone. The additional sources of uncertainty can be identified and added together to create a total uncertainty that can be quite large (e.g. Engel et al., 2008). But the offsetting of individual trace gases is a method to identify systematic errors that can often be attributed to a specific issue, such as the $SF_6$ calibration during
the SOLVE mission in the 1990s. The offset optimization accounts for the calibration issue while still calculating mean ages that agree well with values from missions throughout the 1990s. Without the offset method, measurements from SOLVE, for instance, would not be usable in the optimization because there would be no overlap between solutions for $SF_6$ and $CO_2$. For other missions and locations there may be a minimal set of overlapping solutions due to an inaccurate assumption of photochemical loss for instance. As long as one of the
trace gases, in this case $CO_2$, requires negligible scaling across essentially all the available measurements then the scaling method can identify and account for systematic errors in the measurements or certain aspects of the age calculation. It may be asked why $SF_6$ is used at all from the 1990s if the offset values can be large. The benefits of adding $SF_6$ are many, but primarily it constrains the younger mean ages ($\Gamma < 3$ years) that are not well constrained by $CO_2$, as shown by the example in Figs. 3 and S8, which then allows better constraint on the ratio of moments.

We demonstrate the technique on the extensive *in situ* measurements from the 1990s and the more limited recent measurements in the 2020s. The 1990s average distributions of mean ages, age spectra width and surface source latitudes from the upper troposphere to the lower stratosphere and nearly pole to pole provide a unique view of average transport at that time. The optimized mean ages calculated here are compared to the archived values
previously calculated from $CO_2$ and $SF_6$ measurements independently. In many regions the optimized mean ages are similar to the archived values from $CO_2$, accounting for the surface vs. tropopause source difference. An additional factor that could contribute to older mean ages is the variable surface source latitudes that often result in larger boundary condition $CO_2$ mixing ratios compared to the Mauna Loa-Samoa average. The optimized mean ages are younger than the previously calculated values from $SF_6$ everywhere but in the tropics and SH due to the $SF_6$
mesospheric loss and mean age bias that was not previously accounted for.

The age spectra ratios of moments and surface source latitude distributions from both time periods are unique contributions from this study. The ratios of moments have not been well constrained by observations and the distributions shown here generally agree well with recently published values from CCM output. The surface source



latitudes have a number of interesting features that are beyond the scope of this study to discuss but demonstrate the potential information available from this technique. We also leave comparisons of the results between the two time periods to a follow up study.

      The recent aircraft missions and more systematic balloon flights in the stratosphere provided the primary motivation
for this study, yet this new data is limited in its ability to detect decadal changes in the stratospheric circulation due to the NH-only sampling. It has been known for quite some time that the average difference in mean age between the tropics and extratropics on an isentrope can reveal the strength of the vertical component of the Brewer-Dobson circulation (BDC) (Neu and Plumb, 1999; Linz et al., 2017). Yet the last *in situ* measurements in the tropics above 550 K of the four trace gases used in this study to calculate mean ages were made in the 1990s. The result of this
lack of tropical measurements is that we have to rely on mean age trends from the somewhat more extensive *in situ* measurements in the NH to infer multi-decadal trends in the BDC (e.g. Engel et al., 2008; 2017). This single hemisphere view is an incomplete picture, partly due to hemispheric shifts in the BDC that can occur on decadal time scales (e.g. Ploeger and Garny, 2022). Satellite measurements of long-lived trace gases can be used to calculate age of air (e.g. Haenel et al., 2015) but thus far no satellite has measured two age tracers such as $CO_2$ and $SF_6$
necessary to perform the optimization described here. Thus, *in situ* measurements provide an important complement to satellite measurements and more consistent *in situ* sampling of the tropical and extratropical stratosphere (e.g. Moore et al., 2014) is necessary to better understand and model the ongoing stratospheric circulation changes.

**Competing Interests**


      The contact author has declared that none of the authors has any competing interests.

**Acknowledgements**

This research was supported in part by NOAA cooperative agreement NA22OAR4320151. An International Space Science Institute (ISSI) project focused on stratospheric age of air led by Hella Garny was helpful in the formulation of this work. The NOAA AirCore program is partially funded by NOAA's Earth's Radiation Budget Initiative, NOAA CPO Climate and CI (grant no. 03-01-07-001). Jack Higgs, Timothy Newberger, Sonja Wolter supported routine AirCore-based flights and analysis for this work. The authors declare no conflicts of interest related to this
study.

**Data Availability Statement**

      The processed data supporting this study are available from https://csl.noaa.gov/groups/csl8/modeldata.


**Software Availability Statement**



The IDL software used to perform the data analysis and make the figures in this study are available from
https://csl.noaa.gov/groups/csl8/modeldata.


**Author contribution**

ER designed and carried out the calculations and wrote the manuscript. FM provided measurement data and
conceptual support of the study. HG provided model output and conceptual support. EH, BH, GD, DN, JE, SW, JP,
BD, BB, JL and CS provided measurement data.

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
