# Peer review of "Age of air from in situ trace gas measurements: Insights from a new technique"

_EGUsphere, 2024_

## Author Comment (AC1)

We thank the reviewers for their thoughtful comments. Author responses to each reviewer comment are included below in the red font.

**Reviewer 1**

This manuscript presents a new technique to calculate the age of air from measurements of trace gases. Having better (more accurate) constraints of age of air is a very important topic, especially given the possibility the stratospheric circulation and age of air are changing due to climate change and/or ozone depletion and recovery. The paper is well written (although it did take me a while to understand the details of the method) and I think publishable with only some minor revisions (on the results section).

**Specific Comments:**

Line 470: The authors state that "the 'wing plot' of  $\Gamma$  in the 18-20km altitude range" has been extensively used for model comparisons. So why is this plot not shown for the new calculations?

**This is a good point, we have added a version of the wing plot as the new Figure 6 along with a couple of sentences describing it briefly at the beginning of Section 4.1.**

Fig 6 This is not apples to apples comparison given the different reference location of the age calculations. Can you subtract a characteristic surface to tropical tropopause age (e.g. 3 months quoted on line 489) to make this a cleaner comparison.

This is a valid point. The previously published mean ages shown in Figure 6 (now 7) use a time offset from the surface mixing ratio time series to approximate a tropical tropopause origin rather than the surface origin used in this study. This is discussed further later in the section in regards to Figure 9 (now 10). The issue with subtracting a single time from the mean ages found in this study is that the previously published mean ages assume different transport times from the surface to the tropical tropopause depending on the species and even the mission. While it is true that subtracting 3 months from the ages found in this study would make them closer to agreeing with the previously published CO2 mean ages, the point of this figure is more to clarify that the mean ages found in this study are different from those published previously both because they are from the surface and they account for SF6 loss. So, we would rather not adjust our mean ages in this figure but we do add a sentence following the figure to explain this choice.

Line 610 "A follow up study will examine the differences between the 1990s and 2020s ages in detail". I think there needs to be at least a brief discussion of this issue. You show figures for the two periods, so if you don't comment on this you will leave it up to the reader to draw their own conclusions. Better that you say what your data shows. This doesn't prevent a more detailed analysis.

We have added a brief discussion of the differences between the 1990s and 2020s mean ages.

Fig 10 Can you identify the season of DCOTSS and SABRE, so can see straight away which of the air core they should be compared with. I would also suggest removing the annual mean for air core, as figure is cluttered. ..

**Done.**

Data Availability: The data is available but I am not idlsave files are the best way to archive the data. Why not save as ascii or netcdf files that can be easily read by any software. I think there will be a lot of interest in these data (e.g. for model comparison) and I think it is in the authors interest to present the data in the simplest form for others to use.

**We have now included netcdf versions of the aircraft and balloon mean ages, ratio of moments and surface latitude sources in the archive.**

**Reviewer 2**

Ray et al. present a new technique to calculate both the first (mean age of air) and second (ratio of moments, width of age spectra) moments of age by using in-situ measurements of multiple long-lived tracers. This method uses transit times from the Earth's surface, instead of the commonly used assumption of the tropical tropopause, to any location in the atmosphere. Having an accurate constraint on stratospheric age is especially important to better understand possible shifts in the Brewer-Dobson Circulation in response to a changing climate and its further implications on radiation, chemistry and dynamics.

The main conclusions of this study are: (1) This work presents age derivations using simultaneous in-situ observations of  $SF_6$  and  $CO_2$  from both 1990s and 2020s, which could help address ongoing questions in the field. (2) These derivations were done in the upper troposphere and throughout the stratosphere. In addition, (3) ratio of moments derived from in-situ observations agree well with recently published results from chemistry-climate model output, and lastly, (4) results from records spanning multiple decades will allow for future age comparisons.

Overall, this paper was well-written and concise. The research question(s) and descriptions addressing the science goals were clearly stated. However, the methods section could be written more clearly, which will be further explained in the comments section. Given that this manuscript introduces new methods to better constrain stratospheric age, I believe this manuscript is publishable and of interest to prospective readers in the field and ACP. Below are some suggestions for authors to consider for minor revisions before submission. I am very much looking forward to the follow-up study.

**Technical Comments:**

Equation 2: Past literature used the same assumption that "age spectra are assumed to have an inverse Gaussian functional form." Given that gamma, spectrum width, and ROM are provided, are there any other assumptions made when deriving the result?

No other assumptions are made in the form of the age spectra. We have used the same formulation as many previous studies since this has been shown to be a good approximation of the average age spectra shape.

Line 195: "The fraction f has an age dependence with f(t' < t'i)=0, f(t' > t'f)=1 and an exponential form between t'i and t'f, which are the transition ages between the latitudinally varying and purely tropical source regions."

I wanted to be clear about the notation described in this sentence: t'i is the transit times of the latitudinally varying source region and t'f is the purely tropical source region?

Yes, for ages or transit times younger than t'i the spectrum is composed entirely of the latitudinally varying source region and for ages or transit times older than t'f the spectrum is composed of the purely tropical average surface source. In between these times is the transition between them as shown in Fig. S2. We have added a sentence to the caption of Fig. S2 to include the values of t'i and t'f.

**Specific Comments:**

Line 156: Including a brief one or two sentence description of what the age spectrum is (after introducing Equation 2) would be useful. In reference to this study, it is a mass weighted function generated by different pathways (or colored lines) to the sample region. This would help bridge Equations 1 & 2 and Figure 1.

A sentence has been added here with a brief description of what the age spectrum represents.

Line 233-234: Adding a brief sentence explaining the reasoning of the latitude-season gradients in  $CO_2$  (and not  $SF_6$ ) would provide more context when presenting results in Figure 2.

We have added a sentence here briefly describing the variability of the surface gradients of CO2 and SF6 and reference to a NOAA GML web site that shows the CO2 latitudinal and seasonal surface gradients.

Line 340: Include a reference to Figure S5 in the caption of Figure 3 so the reader can refer back to labels of the dashed lines in supplemental.

**Done**

Line 470: There is a reference to a 'wing plot' in the manuscript and Age-N2O relationships, where there are Age-N2O results from this study. Given that there are zonal mean ages as a function of height and latitude derived from both flight and balloon data in this work and that  $CO_2$ -age and  $SF_6$ -age have been "been used extensively" with the wing plots, why was this comparison not done in this study?

A version of the wing plot has now been added as the new Figure 6.

Figure 6: Which latitude ranges were used for both - balloon and aircraft - relationships from 1) this study, 2) Andrews et al. and 3) Volk et al., and are they the same? Tracer interrelationships vary with latitude as a result of the overturning circulation, and in the case of SF6, mesospheric influence/ sinks. For example, one would expect the tropics Age-N2O relationship to have younger air at almost all normalized N2O bins compared to that of Age-N2O at 60N where air has been in the stratosphere longer and has been subjected to mixing/ sinks. This is important to include in both the text and figure description and for comparing Age-N2O relationships from different studies.

The latitude, altitude and seasonal sampling are essentially the same for each of the sampling platforms, so for each plot in the figure the sampling is the same for each of the correlations. We've added a couple of sentences describing the sampling. Of course,  $SF_6$  has much reduced sampling compared to  $CO_2$  so there are fewer measurements represented in those correlations.

It is true that tracer interrelationships often vary with latitude, altitude and season. This is demonstrated for example by the figure below that shows mean age vs. N2O from a WACCM model run. This figure includes all model levels and shows three separate latitude regions over the 1990s. There is a difference in mean age with latitude although it is most noticeable for N2O

The next figure shows how the correlation range and average change when only considering altitudes, latitudes and months sampled by the aircraft. The blue line is the model equivalent of the sampled correlations in Figure 6a. This shows how compact the correlation is with the sampling restrictions and validates the compactness of the correlations shown in the paper. A discussion of these correlations and what causes them to vary could be the subject of its own

paper and they have in fact been discussed in previous work going back to for instance Plumb and Ko, 1992.

While your points about the variability in the correlations are valid it is beyond the scope of this paper to include a discussion about this topic. We include the figures in Section 4 mostly as a demonstration of the mean ages calculated here in the context that they have been shown in previous studies but not as a comprehensive comparison.

Figure 7 and 8: the Age color bar ranges are slightly different for Figures 7a and 8a. The scaling ranges and pixel spacing should be included in the figure descriptions. (This can also be applied in description of Figure 11 as well)

The age color bar ranges were intentionally changed in Figs. 7, 8 and 11 in order to highlight the features in the different periods and sampling ranges. Notes in the captions have been included in each of the figures to indicate the grid spacing.

Section 4.2: Everything was well-written and concise, although a bit short. I see that "a followup study will examine the differences between the 1990s and 2020s ages in detail," but I think stating what the results show for the reader will not take away from the follow-up study. For example: there is a seasonal shift in AirCore relationships- why?

We have added a couple of sentences at the beginning of this section briefly describing the differences between the 2020s and 1990s correlations, also in response to another reviewer comment. The seasonal variability seen in the AirCore correlation is interesting and generally consistent with the WACCM model output shown in the plots above. But again, this would

require some extended discussion to describe that is beyond the scope of this paper mostly because there is no reference to cite or established expectations. The AirCore results will be the subject of a follow up paper that will bring in model output and more fully describe the seasonal and potential QBO variability seen in the measurements and model.

Figure 10: Similar to Figure 6, what latitude ranges were used for the Age-N2O relationships and are they similar enough to be comparable. Including the latitude ranges in the figure descriptions would be useful.

In the second paragraph of this section while describing Fig. 10 there is a mention of the latitude and theta ranges of each mission shown in Fig. 11. We have added a sentence in the caption to see Fig. 11 for the sampled ranges included in the data shown in Fig. 10.

**Reviewer 3**

This is an interesting study in determining several statistical terms related to age-of-air, including some terms that have not previously been shown in other studies. It also shows nicely how the use of two tracers can be used to better constrain the age-of-air. I apologize for my lack of expertise in evaluating this study, but hopefully I can provide suggestions that help to clarify some concepts for a wider audience.

Please explain why, in equation (3), G is partitioned into a tropical surface source and a latitudinally varying surface source (which seems to include the tropics). I certainly understand that it is useful to show the tropical term (e.g. the dashed line in Figure 2), but couldn't one simply determined this by letting gV indicate the complete latitudinally varying surface source and then summing up gV from 30S-30N to find the tropical contribution? Maybe this is all explained in a previous study, but some explanation here would be helpful.

We have added a couple of sentences before Eq. 3 to briefly explain the reason we partition the age spectra in this way. The primary reason is to account for transport of air directly from the extratropical surface to the UTLS region. This can have a significant effect on the UTLS composition of trace gases with large latitudinal surface gradients such as SF6 and CO2. Without accounting for the possibility of this extratropical surface influence, age of air estimates can be biased.

Another way to do this calculation is to do as you say and use the full age spectra G in the way that we use gV. That is, we could convolve the age spectra for all ages with a range of surface latitudes to see which latitudes fit the data best. This would likely yield similar results to what we show here but we have not done the calculation to confirm that. The reason we have partitioned the age spectra to have a latitudinally varying portion, that does include the tropics and extratropics, and a tropical only portion is based on our previous Ray et al., 2022 study where we could clearly identify the influence of short time scale transport from the extratropical surface to the UTLS with this technique. Although that is not necessarily the focus

of this study, we still considered this aspect of the age of air technique that can add transport information to be useful. There are certainly many ways to do the age of air calculation and we only intend to demonstrate the technique we have chosen.

The term on the right-hand side of (3) is "non-normalized", while the terms on the left-hand side are "normalized". I am not sure what exactly this means, but I do not understand how adding two non-normalized terms can produce a normalized term.

We start with the normalized full age spectra G and then partition it into the two parts based on the function f that is described in the following paragraph and Eqs. 4 and 5. That ensures the total age spectra are always normalized while the two parts are not.

The notation y\_oTR in (7) is confusing. I don't think this is used elsewhere, and the presence of a surface source latitude parameter on one side of this equation and not on the other seems problematic.

This has been clarified by adding a reference to y\_oTR in the sentence following Eqs 7 and 8 to indicate that it refers to the surface tropical average (30S-30N) latitude. We have also added a sentence in this paragraph explicitly stating that Chi\_iTR should not have a surface latitudinal dependence since y\_oTR represents a single tropical latitudinal average.

In consideration of this comment, we have also changed the notation in Eq. 6 from y to  $y_0$  and in Eq. 8 from  $y_0V$  to  $y_0$  on both the left and right sides, and all subsequent references in the text of  $y_0V$  to  $y_0$ . We have also explicitly stated that the subscript o refers to a surface quantity. We hope that this clarifies the surface latitudinal dependence of the parameters.

Figure 4 – the polaris and solve colors are very similar and difficult to distinguish. Also, please give some indication of the altitudes or pressure that are being shown here. Of particular interest is how much of this data is in the stratosphere?

The SOLVE symbol colors were changed. And a sentence has been added in the caption to clarify that essentially all of the data shown in this figure are from the stratosphere (where normalized N2O < 1) at altitudes up to ~32 km. This is made clear in Section 4 but it is useful to point that out here.

Line 405 - I think this is the first time the subscript 's' is used in the text. If so, please define it here. I did eventually find the definition in the Figure 4 caption.

**Done**

Paragraph starting at line 430 – I think the authors are saying that there was an SF6 measurement problem during SOLVE. If this is correct please state so clearly. If not, please clarify the paragraph to explain the problem.

A sentence has been added at the end of the paragraph clearly stating that there was a high bias in the SF6 measured on the ER-2 during SOLVE.

---

## Author Response (AR2)

All of the editor's comments have been implemented.  Thank you for your careful reading of the manuscript.